# The ratio of shock index to pulse oxygen saturation predicting mortality of emergency trauma patients

**Junfang Qi**[1☯], **Li Ding**[1☯], **Long Bao**[1], **Du Chen**[2]*

1 Department of Emergency Medicine, the First Affiliated Hospital of Soochow University, Suzhou, China,
2 Department of Critical Care Medicine, the First Affiliated Hospital of Soochow University, Suzhou, China

☯ These authors contributed equally to this work.
* sdfyycd@suda.edu.cn

## Abstract

### Objective

To test the following hypothesis: the ratio of shock index to pulse oxygen saturation can better predict the mortality of emergency trauma patients than shock index.

### Methods

1723 Patients of trauma admitted to the Emergency Department of the First Affiliated Hospital of Soochow University from 1 November 2016 to 30 November 2019 were retrospectively evaluated. We defined SS as the ratio of SI to SPO2, and the mortality of trauma patients in the emergency department as end-point of outcome. We calculated the crude HR of SS and adjusted HR with the adjustment for risk factors including sex, age, revised trauma score (RTS) by Cox regression model. ROC curve analyses were performed to compare the area under the curve (AUC) of SS and SI.

### Results

The crude HR of SS was: 4.31, 95%CI (2.89–6.42) and adjusted HR: 3.01, 95%CI(1.86–4.88); ROC curve analyses showed that AUC of SS was higher than that of shock index (SI), and the difference was statistically significant: 0.69, 95%CI(0.55–0.83) vs 0.65, 95%CI(0.51–0.79), P = 0.001.

### Conclusion

The ratio of shock index to pulse oxygen saturation is good predictor for emergency trauma patients, which has a better prognostic value than shock index.

**Data Availability Statement:** All relevant data are within the manuscript and its Supporting Information files.

**Funding:** The authors received no specific funding for this work.

**Competing interests:** The authors have declared that no competing interests exist.

## Introduction

Trauma is a worldwide public health issue, causing serious economic and medical burdens [1]. Trauma patients have many symptoms, serious injuries, rapid and changeable disease progress, and there is a risk of death at any time. It is reported that there are three obvious death peaks in trauma patients,: they occur within 1 hour (about 50%), 3 hours (about 30%), and 1–4 weeks (about 15%), respectively after injury. The first two death peaks with the highest proportion occurred within a few hours in the early stage of trauma. For this reason, it is very important to reduce the early mortality of trauma, which requires that doctors could quickly predict early mortality and identify trauma patients at risk of early death. The prediction of early mortality has important guiding significance for activating the trauma team, preparing for surgery as soon as possible and good communication between doctors and patients. Therefore, the aim of our study is to seek for a handy and rapid method that can quickly predict early mortality of trauma patiens in ermengency department (ED).

The vital signs are first-hand information that we could obtain in the ED.The shock index (SI), the ratio of heart rate to systolic pressure, can be easily calculated depending on vital signs and has been proven as a good predictor in clinical practice [2]. There was a study that indicated that the elevated SI recorded in the ED increased the probability of both hospital admission and inpatient mortality in the general adult ED population [2]. Studies suggested that SI could predict the prognosis of trauma patients [3–5]. The increase of SI mainly indicated acute hypovolemia and circulatory failure among trauma population [6] and it significantly correlated with the days of hospitalization, intensive care, mechanical ventilation and the risk of mortality [3].

The literature on the relationship between pulse oxygen saturatrion (SpO$_2$) and the prognosis of trauma patients is limited and has the inconsistent conclusion. A study reported that SpO$_2$ do not add significant value to other variables when predicting mortality in severe trauma patients [7]. However, some researches believed that SPO$_2$ can effectively predict the early mortality of trauma patient [1, 8]. These results demonstrate a need to further evaluate how vital signs affect mortality of trauma patients.

Considering the rapid availability of SI and SpO$_2$ and previous studies suggesting the predictive value of SI and SpO$_2$ for the prognosis of trauma patients, this study proposed a new index SS based on the ratio of SI to SpO$_2$ as a predictor for mortality of trauma patients in ED and evaluated whether the SS can better predict the prognosis of emergency trauma patients than SI.

## Methods

### Study design and participants

This was a retrospective study using data from the emengency trauma registry information system ED of the First Affiliated Hospital of Soochow University. Data from 1 November 2016 to 30 November 2019 were export from the database. Patients' characteristcs, including age, sex, revised trauma score (RST), shock index (SI), pulse oxygen saturation (SpO$_2$), shock index/SpO$_2$, mean arterial pressure (MAP), pulse, respiratory rate (RR), body temperature (T), as well as mortality in ED, were recorded in the dataset. These vital signs data such as SpO2, MAP, RR, T, etc.used in this study were collected by the first measurement when the trauma patients have just entered the emergency department and have not been treated in the emergency department. Informed consent was not rquired because the data were collected without identifiable personal information. The inclusion criteria were: 1. patients categorized with blunt or penetrating mechanisms; 2. Age ≥18 years. The exclusion criteria consisted of: 1.

Age<18 years, pregnant women; 2. Patients who died at the time of admission or voluntarily gave up treatment; 3. Patients with incomplete information that the study required. The data of our study were taken from the ememgency trauma registry information system of our hospital and all data of patient were collected without identifiable personal information. The study was approved by the Ethics Committees of the First Affiliated Hospital of Soochow University (Suzhou, China) and the Institutional Review Boards (the Ethics Committees of the First Affiliated Hospital of Soochow University) waived the need for informed consent before analysis due to the retrospective nature of the data. This study conforms to the principles outlined in the Declaration of Helsinki.

## SS

Because the vital signs are systemic, associations between individual vital sign and trauma patients' mortality may not be always apparent. Hence, combining SI with $SpO_2$ may provide a more accurate predicton of mortality of trauma patients. Therefore, we proposed the concept of SS ($SI/SpO_2$). We hypothesized that SS has a better performance in mortality predicting of emergency trauma patients than shock index.

## Prognosis evaluation

The end-point of outcome was mortality of trauma patients in the ED, which was timely recorded by emergency doctor in trauma registry. According to whether the patients survived or not, patients were dichotomized into two groups: survival and non-survival. The correlation between SS and mortality in ED was analyzed.

## Statistical analysis

Continuous variables were tested for normality using Shapiro–Wilk test. All of the continuous variables in the current study, failing to conform to normality, were thus expressed as median (inter quartile range, IQR) and compared using Mann-Whitney test. Categorical variables were expressed as frequencies and percentages and compared using Likelihood-ratio Chi squared test. Cox regressions were performed to caculate the hazard ratios (HRs) of variables for death. Model 1: crude HRs were reported with no adjustment for each risk factors; model 2: adjusted HRs, with the adjustment for risk factors including sex, age, revised trauma score. Receiving operating characteristic curve analyses were performed to define the cutoff values of variables for discriminating between survival and non-survival. Statistical analyses and graphics were completed with STATA 15. Two-tailed P<0.05 was considered to be statistically significant.

## Results

A total of 1723 patients were evaluated in the study, including 1259(73.07%) males and 464 (26.93%) females, 1692 (98.20%) in the survival group and 31 (1.80%) in the non-survival group. The retention time of non-survival group in ED was 1–232 hours, with a median of 17 hours. There were significant differences between two groups in RTS, SI, $SPO_2$, SS, MAP, T, RT (P<0.05). SI and SS of the non-survival group were significantly higher than that of the survival group (P = 0.004 and P<0.001, respectively), while RTS, SPO2, MAP and T were significantly lower than that of the survival group (Table 1).

 Univariate COX regression analysis revealed that the mortality of emergency trauma patients was closely related to RTS and SS (P < 0.001). The crude HR of SS was 4.31, 95%CI (2.89–6.42). Multivariate COX regression model of sex, age, RTS and SS identified that SS was

**Table 1. Baseline characteristics.**

| Variables | Survival 1692(98.20) | Non-survival 31(1.80) | P value |
|---|---|---|---|
| **Sex (n, %)** | | | 0.887 |
| **Female** | 456(26.95) | 8(25.81) | |
| **Male** | 1236(73.05) | 23(74.19) | |
| **Age (year)** | 51(25) | 50(19) | 0.925 |
| **RTS** | 12(0) | 10(0) | <0.001 |
| **SI** | 0.64(0.24) | 0.98(0.85) | 0.004 |
| **SpO$_2$ (%)** | 98(4) | 91(15) | <0.001 |
| **SS** | 0.65(0.26) | 1.19(0.97) | <0.001 |
| **MAP (mmHg)** | 99(22) | 77(53) | 0.016 |
| **P (n/min)** | 85(23) | 98(50) | 0.065 |
| **RR (n/min)** | 20(4) | 20(13) | 0.068 |
| **T (˚C)** | 36.9(0.8) | 36.0(1.9) | <0.001 |
| **RT (hours)** | 4(13) | 17(35) | <0.001 |

Continuous variables were expressed as median (IQR); categorical variables were expressed as n/percentage; P values were caculated by Mann-Whitney test. RTS, revised trauma score; SI, shock index; SpO$_2$, pulse oxygen saturation; SS, shock index/SpO$_2$; MAP, mean arterial pressure; P, pulse; RR, respiratory rate; T, body temperature; RT, retention time in the ED.

an independent mortality predictor of emergency trauma patients. The adjuted HR of SS was 3.01, 95%CI(1.86–4.88), suggesting that with one unit increased of SS, the risk of mortality was raised by 2.01 times (Table 2).

The ROC curve analyses showed the area under the curve (AUC) of SS was 0.69, 95%CI (0.55–0.83), which was higher than the that of SI (P<0.001) (Table 3). As shown in the ROC curve (Fig 1), each point on the curve corresponded to a sensitivity and specificity, and a high sensitivity meant a decrease in specificity. In order to determine an ideal cutoff value, we used the Youdenindex method (Youdenindex = sensitivity + specificity-1), which meant that we used the SS value corresponding to the maximum value of Youdenindex as its cutoff value. The cutoff value of SS was 1.06 (sensitivity: 92.26%, specificity: 61.29%, Youdenindex:0.5355) (Fig 1).

## Discussion

Our study identified SS as an independent mortality predictor of trauma patients in the ED. It could be a better choice than shock index for assessment and triage of trauma patients in the ED.

**Table 2. Cox regression analyses.**

| Variables | Univariable | | Multivariable | |
|---|---|---|---|---|
| | HR(95%CI) | P value | HR(95%CI) | P value |
| **Male** | 1.01(0.45,2.27) | 0.982 | 0.75(0.25,2.25) | 0.613 |
| **Age (year)** | 1.00(0.98,1.02) | 0.909 | 1.01(0.98,1.04) | 0.459 |
| **RTS** | 0.63(0.57,0.70) | <0.001 | 0.64(0.56,0.73) | <0.001 |
| **SS** | 4.31(2.89,6.42) | <0.001 | 3.01(1.86,4.88) | <0.001 |

SS, shock index/SpO$_2$, RTS, revised trauma score.

**Table 3. Analyses of receiver operating characteristic curves.**

| Variables | AUC | 95%CI |
|---|---|---|
| SI | 0.65 | (0.51,0.79) |
| SS* | 0.69 | (0.55,0.83) |

SS, shock index/SpO$_2$; SI, shock index; AUC, area under the curve

* Comparision of AUCs: P = 0.001.

The majority of injury-related deaths occur in Low-income and middle-income countries (LMICs), human and technological resources care are constrained in ED [9]. It is crucial to identify traumatized patients at risk of early death. A variety of methods have been developed for this purpose, Including the classic mehods and new methods. The former include the Injury Severity Score (ISS) [10], the Revised Trauma Score (RTS) [11], the Trauma and Injury Severity Score (TRISS) [12] and the latter consists of the MGAP score [13], the GAP score [14], the New Trauma Score (NTS) [15].

The ISS is an atomical score consisting of Abbreviated Injury Scale codes for the three most severely injured body regions [9]. The RTS is a physiological scoring system consisting of Glasgow Coma Scale (GCS), systolic blood pressure (SBP) and respiratory rate (RR). Each Parameter is transformed to coded value that is multiplied by a weighted coefficient before it is added [15]. The TRISS is the most classic model, consisting of the ISS, RTS, age, and mechanism of injury. No trauma scoring system performed better than the TRISS in predicting survival probabilities [15]. But, In the absence of coding charts or computers, these trauma scoring systems are impractical because of difficulties in information collection and real-time calculation.

In recent years, some simplified methods were proposed to evaluate the prognosis of patients with trauma. Kimura et al. [16] developed a simplified alternative to the TRISS method that is (coded GCS + coded SBP + coded age–constant), So a coding chart is still Indispensable. The MGAP and MAP are recently developed trauma scoring systems and they are simple, rapid scoring systems in the prediction of trauma-associated mortality. The MGAP utilized mechanism, GCS, age, and arterial pressure and the GAP is a method that is simplified by deleting the mechanism from the MGAP [14]. They can be easily calculated after the simple addition of several numbers and are more accurate than other trauma scoring systems (TSSs). But it is almost impossible to calculate without a scoring chart.

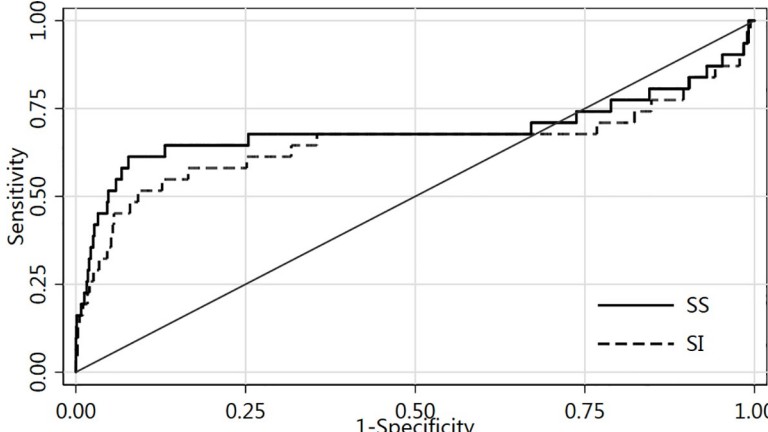

**Fig 1. ROC curves of the SS and SI.**

SI is a classic indicator that is more sensitive than traditional vital signs to evaluate shock. In recent decades, there have been a large number of studies on the practical value of SI in trauma patients. A retrospective cohort study of 16269 trauma patients elucidated the relationship between severe pre-hospital SI and the days of in-hospital, length of stay of ICU, days of mechanical ventilation and use of blood products and suggested that SI > 0. 9 indicated a higher risk of transfer to ICU, emergency surgery or death [3]. A research of 1419 patiens indicated that patients who continued to receive high SI after 1L Crystal liquid resuscitation had a higher demand for blood transfusion and higher mortality and worse outcomes [17]. This conclusion was confirmed in a study [18] that showed if SI did not improve within six hours, there would be a significant increase in mortality. A recent finding reported that abnormally elevated SI at any time point suggests that trauma patients have a higher risk of dying within 28 days [19]. Charry et al. reported that SI > 0.9 predicted a worse prognosis after trauma [20]. These previous studies all highlight a finding that there is a statistically significant correlation between SI and mortality in trauma patients. It is indisputable that SI is often used as an indicator of severity and poor prognosis in trauma patients, and its abnormally elevated levels often indicate a worse outcome in trauma patients. However, most of these studies on the relationship between SI and the mortality of trauma patients are focused on the mortality of trauma patients during hospitalization or 28 days. It's not the early mortality during the stay in the ED.

At present, the study on the relationship between SPO2 and the prognosis of trauma patients is limited. A retrospective study indicated that RR and $SpO_2$ do not add significant value to other variables in the RTS and TRISS when predicting mortality in severe trauma patients [7]. However, the author said this research might be insufficient to detect significance due of sufficient missing information. On the contrary, A prospective study [15] showed that $SpO_2$ was a better parameter than RR and could add significant value to other variables in predicting mortality in trauma patients. Otherwise, A recent study revealed pragmatic value of $SpO_2$ in trauma and indicated that SPO2 can effectively predict the 24-hour mortality of trauma patients [1].

The vast majority of the previous studies on the prediction of early in trauma patients are based on the mortality in-hospital or 28 days [3, 6, 9, 19, 21, 22], not early mortality in the emergency room. A recent study [1] Identified variables that can be quickly measured to predict 24-hour early mortality, showing that SPO2 < 90% can be used to predict early mortality. However, it did not address the correlation between SI and early mortality in trauma patients.

As mentioned earlier, these points about SI and $SpO_2$ led us to investigate whether the ratio of SI to $SpO_2$ (SS) is a better predictor of early mortality in the ED and whether the SS could better predict the prognosis of emergency trauma patients than SI. To the best of our knowledge, this is the first study to show that SS is a practical tool for identifying high-risk trauma patients. SS is easily available in ED without the need for additional imaging, hard-to-remember charts, or equipment. Due to real-time calculation, SS can be utilized as a realistic and satisfactory tool for real-time evaluation, reasonably allocating medical resources in ED. In addition to being convenient for calculation, this study showed that the AUC of SS is higher than that of SI and the difference is statistically significant (P = 0.001). The larger the AUC value, the better the predictive or diagnostic value of the index. Therefore this finding indicated that SS had better discriminant ability for mortality risk than SI. Besides, It could serve as as a triage tool for trauma victims in large-scale trauma events, which allowed patients with serious injuries to be quickly sorted at the entrance upon arrival.

Compared with SI, our subjects SS included SpO2 and the reasons were as follows. It was reported that indicators corelated with microcirculation perfusion were important predictors of mortality in trauma patients, such as lactate clearance, base deficit and so on [1, 18, 22].

Therefore, it is valuable to include these predictors in the method of predicting mortality in general trauma population. However, these laboratory parameters must take time to measure and they are mostly evaluated in patients with severe injuries and not often tesed in general trauma people, which results that this information is always missing in general trauma patients. So, it is not feasible to include these predictors in the method of early and real-time prediction of mortality in general trauma population in ED. Physiologically, SpO2 could reflect not only microcirculation, but also tissue perfusion and oxygenation [7], and its advantages of real-time and non-invasive monitoring facilitate timely evaluation in ED. Taking these into account, it is appropriate to include SpO2 in our study on the prediction of mortality among trauma patients in ED. Furthermore, hemorrhage and traumatic brain injury (TBI) are the main causes of early death in trauma patients [1]. Shock index (SI) is considered to be a reliable indicator of early hemorrhage and can predict the prognosis of trauma patients [2], but when acute traumatic brain injury (TBI) is complicated with hemorrhage in trauma patients, the performance of SI has not been reported. There was only one animal experimental study [23] which indicated that SI is not a reliable indicator of progressive bleeding after moderate TBI and it seriously underestimates the potential bleeding risk of moderate acute TBI. On the contrary, the control group and mild TBI group, there was a good linear relationship between SI and progressive bleeding. Considering this, it is biased to only use SI to evaluate the prognosis of all traumatic patients. After all, TBI accounts for a certain proportion in trauma. As far as we know, a study conducted by Sobuwa have concluded that SpO2 is independently significant predictors of outcome in severe TBI. The investigation showed that patients with SpO2 ≥ 90% were more likely to get good results (214.8%) [24]. Considering the deficiency of SI and the value of SpO2 in non-mild TBI patients, it seems necessary to combine these parameters. In traumatic people, the shock is mostly hypovolemic shock caused by blood loss, which is still associated with significant mortality [19]. when shock patients have decreased SpO2 at the same time, it is mostly due to the loss of too much hemoglobin, decreased oxygen-carrying capacity of the body, and tissue ischemia and hypoxia, which often suggests that the condition is more critical and requires early attention and intervention to replenish the blood volume while opening the airway to increase oxygenation. Especially when complicated with TBI, it is more necessary to detect and treat it in time, because the brain is very vulnerable to ischemic injury and TBI patients will benefit if hypotension and hypoxemia can be avoided as the situation [24]. Due to very low oxygenation or poor peripheral circulation, SpO2 may not be measured in some situation where patients might suffer from severe hemorrhagic shock, tension pneumothorax, cardiac tamponade and so on [7]. That failing to test SpO2 in trauma patients needs more cautions, which might indicate that patients are in a urgent and life-threatening state and suggest that we should actively take rescue to maintain vital signs, rapidly identify the causes of poor oxygenation at the same time, and remove the risk factors in time. As above, SS that Combine SI with SpO2 indeed a valuable and practical tool in ED.

In addition, the mortality of trauma patients in our study was lower than that of other studies, which may be due to the fact that we excluded patients who died in pre-hospital or arrived at the emergency room without vital signs and patients who automatically gave up treatment. They were usually in critical conditions where the treatment cost was too high and the benefit was too small, and some families will give up. Some previous studies were limited to multiple injuries [18], penetrating injuries [20], traumatic Brain Injury [25] and so on, which reseult in a higher mortality. Hence, there is a lower mortality rate in our study than previous studies.

Our study firstly combine SI and SPO$_2$ to evaluate the prognosis of trauma patients, but it also possesses some limitations: First, it was a single-center retrospective study, so more multi-center prospective studies were needed to explore the predictive value of SS in the prognosis of

trauma patients. Second, patients with incomplete information were excluded, so the results may be compromised. Third, the vast majority of patients were blunt injuries, this finding may not be applicable to penetrating injuries, especially gunshot injuries.

In conclusion, our study indicated that SS (SI/SpO$_2$) is a practical predictor for emergency trauma patients, and it has better predictive value than shock index. SS can be used to evaluate and triage trauma patients, which is helpful for reasonably allocating medical resources in the ED.

## Supporting information

**S1 Checklist. STROBE statement—checklist of items that should be included in reports of observational studies.**
(DOCX)

**S1 Data.**
(CSV)

## Acknowledgments

We would like to acknowledge the First Affiliated Hospital of Soochow University for providing database that supported this research.

## Author Contributions

**Conceptualization:** Li Ding, Long Bao, Du Chen.

**Data curation:** Du Chen.

**Formal analysis:** Du Chen.

**Investigation:** Li Ding.

**Methodology:** Li Ding, Du Chen.

**Writing – original draft:** Junfang Qi.

**Writing – review & editing:** Junfang Qi, Li Ding, Long Bao, Du Chen.

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
