## [Decision Letter · Decision Letter 0]

25 May 2020

PONE-D-20-08124

The ratio of shock index to pulse oxygen saturation predicting mortality of emergency trauma patients

PLOS ONE

Dear Dr. Chen,

Thank you for submitting your manuscript to PLOS ONE. After careful consideration, we feel that it has merit but does not fully meet PLOS ONE’s publication criteria as it currently stands. Therefore, we invite you to submit a revised version of the manuscript that addresses the points raised during the review process.

This is a retrospetive study based on many patients.  As the first reviewer commented, there is a difference between patients dying within the first hours and those dying following many days.  There is no differentiation between these two population of patients.  It may be that you cannot perform this differentiation since the median time of death indicated, reveals that most of your patients died within one day.  As the first reviewer commented, it may be that we do not need any index to identify this group of patients.

We look forward to receiving your revised manuscript.

Kind regards,

Itamar Ashkenazi

Academic Editor

PLOS ONE

Journal Requirements:

Reviewers' comments:

Reviewer's Responses to Questions

**Comments to the Author**

1. Is the manuscript technically sound, and do the data support the conclusions?

Reviewer #1: Yes

Reviewer #2: No

2. Has the statistical analysis been performed appropriately and rigorously? 

Reviewer #1: Yes

Reviewer #2: No

3. Have the authors made all data underlying the findings in their manuscript fully available?

Reviewer #1: Yes

Reviewer #2: No

4. Is the manuscript presented in an intelligible fashion and written in standard English?

Reviewer #1: Yes

Reviewer #2: Yes

5. Review Comments to the Author

Reviewer #1: 1. endpoint analysis: the subject of ED demise for SS test is, in my opinion, is less important than 30 day mortality. most acute medicine caregivers will be a able to discern a patient in obvious distress who is in deep shock and who is going to die. It will make more sense to use the SS for judging a longer period prognosis

2.English:

row 26 : trauma and not tauma

29: there is a study that indicated

33: the literature is limited

95: traumatized or trauma

172: gave up

173 : and some families will give up

Reviewer #2: Title: The ratio of shock index to pulse oxygen saturation predicting mortality of emergency trauma patients (PONE-20-08614)

Study type: retrospective study

Authors’ methodology and main findings: The authors evaluated 1723 trauma patients admitted to one hospital during a three year period. The yield of a new variable SS (ratio of shock index to oxygen saturation) was evaluated against shock index alone in predicting mortality. ROC curves are provided. These show that the new index better predicts mortality compared to shock index alone.

Reviewer's comments: Many types of scales exist that allow predicting mortality. As the authors state in the beginning of their study: “Our study identified SS as an independent mortality predictor of trauma patients in the ED. It could be a better choice than shock index for assessment and triage of trauma patients in the ED”. SI is a very crude index and as such, any addition to the model could theoretically improve it. This addition could be oxygen saturation as is indicated by this study. However, this addition could also be respiratory rate, GCS, age, type of trauma center, country where the patient is treated, and many others (even the color of the patients’ eyes). Any variable that is able to differentiate between survival and non-survival in a univariate analysis is a potential candidate that can improve the statistical model. The whole idea behind the shock index is that is a simple alternative to other scores. It is for this reason that I cannot recommend accepting this manuscript for publication. The following issues need to be amended in this manuscript:

1. In the introduction, in the description of their aims, the authors need to explain why is it important to quickly predict early mortality? Are we going to treat these patients in a different way compared to other trauma patients (trauma team activation, for example)? Are we not going to treat these patients?

2. Since shock index and pulse oxygen saturation are dynamic and dependent on different circumstances, they should describe when the data was collected - prehospital, upon arrival to the ED, worse measurements during first 6 hours. There needs to be a consistency. Furthermore, there may be a significant difference on SpO2 at the time of measurement if patients arrived were intubated or not, if they were no oxygen supplementation or on room air.

3. In Table 1, the percentages inside the parenthesis are unclear. It is also unclear what the P values represent. It is easy to comprehend that some kind of t-test was performed for age differences. But what kinds of statistical analysis were done for RTS, SI and SS? Where these taken to be continuous variables? I do not think this is correct. RTS is a score composed of added values of scores given for different variables. An RTS of 12 in survival is not 20% higher compared to RTS of non-survivors. The same is true for SI. Though composed of the relationship of two continuous variables, the relationship of SI to mortality is more categorical. Thus SS is more of a categorical variable than a continuous one.

4. In order for this index to be relevant, it is not enough to present the sensitivities and specificities. One should construct a table 3 with different cutoffs for SS in which the proportion of patients dying and surviving is presented for each cutoff value. The authors of the shock index provide us with an optimal cutoff to which we can related when assessing the patients’ prognosis. Why did the authors of this manuscript only offer us an AUC evaluation?

6. PLOS authors have the option to publish the peer review history of their article (what does this mean?). If published, this will include your full peer review and any attached files.

Reviewer #1: No

Reviewer #2: No

---

## [Author Response · Author response to Decision Letter 0]

26 Jun 2020

First of all, we thank both reviewers and editor for their positive and constructive comments and suggestions.

Replies to Reviewer #1:

Comment 1: endpoint analysis: the subject of ED demise for SS test is, in my opinion, is less important than 30 day mortality. most acute medicine caregivers will be a able to discern a patient in obvious distress who is in deep shock and who is going to die. It will make more sense to use the SS for judging a longer period prognosis.

Response: There are three obvious death peaks in trauma patients, as shown in the following figure: they occur within 1 hour after injury (about 50%), within 3 hours (about 30%), and 1-4 weeks (about 15%). The death mechanism of each peak is different. As you said, it would be a better idea to follow up on the patient's survival at 30 days after injury, so that it could basically cover the three death peaks of trauma patients. Unfortunately, due to the limitations of the retrospective study, we were not able to get 30-day prognosis data in the emergency information system. The emergency information system recorded the survival status of patients during the emergency period, which basically covered the first two death peaks of trauma and the third death peak of some patients. As a result of patients leaving the emergency department later, the third death peak of a small number of patients may not be recorded, thus forming a certain censored data. A better way to deal with the censored data in statistics is to carry out survival analysis. Accordingly, we used the Cox regression model to calculate the crude HR and adjusted HR of SS, which can better reflect the prognostic evaluation value of SS in trauma patients on the basis of the existing data.

图片引自：https://aneskey.com/the-development-of-trauma-systems/

Comment 2.English:

row 26 : trauma and not tauma

29: there is a study that indicated

33: the literature is limited

95: traumatized or trauma

172: gave up

173 : and some families will give up

Response: Thank you very much for your suggestions. I am very sorry for our incorrect writing and I have modified these according to the the comments. Revised portion are marked in red in the paper. 

Special thanks to you for your good comments.

Replies to Reviewer #2:

Comment 1：In the introduction, in the description of their aims, the authors need to explain why is it important to quickly predict early mortality? Are we going to treat these patients in a different way compared to other trauma patients (trauma team activation, for example)? Are we not going to treat these patients?

Response: There are three obvious death peaks in trauma patients, as shown in the following figure: they occur within 1 hour (about 50%), 3 hours (about 30%), and 1-4 weeks (about 15%), respectively after injury. The first two death peaks with the highest proportion occurred within a few hours in the early stage of trauma. For this reason, it is very important to reduce the early mortality of trauma, which is based on the rapid and accurate assessment of injury and prognosis of trauma patients in the early stage. The prediction of early mortality has important guiding significance for activating the trauma team, preparing for surgery as soon as possible and good communication between doctors and patients. Shock index (SI) is a simple and effective index to judge the severity of injury. As you said, SI is simple but rough, so we try to introduce another clinically easily available and important indicator SPO2 to get SS, the study shows that SS is also simple to calculate and its prognostic value is better than SI, which has a certain clinical application value.

图片引自：https://aneskey.com/the-development-of-trauma-systems/

Comment 2: Since shock index and pulse oxygen saturation are dynamic and dependent on different circumstances, they should describe when the data was collected - prehospital, upon arrival to the ED, worse measurements during first 6 hours. There needs to be a consistency. Furthermore, there may be a significant difference on SpO2 at the time of measurement if patients arrived were intubated or not, if they were no oxygen supplementation or on room air.

Response: As you said, the consistency of data collection is very important. The vital signs data such as SpO2 used in this study were collected by the first measurement when the trauma patients have just entered the emergency room and have not been treated in the emergency room. Considering the Reviewer’s suggestion, we have made a supplement on the time of data collection in Study design and participants(第几行).

Comment 3：In Table 1, the percentages inside the parenthesis are unclear. It is also unclear what the P values represent. It is easy to comprehend that some kind of t-test was performed for age differences. But what kinds of statistical analysis were done for RTS, SI and SS? Where these taken to be continuous variables? I do not think this is correct. RTS is a score composed of added values of scores given for different variables. An RTS of 12 in survival is not 20% higher compared to RTS of non-survivors. The same is true for SI. Though composed of the relationship of two continuous variables, the relationship of SI to mortality is more categorical. Thus SS is more of a categorical variable than a continuous one.

Response: In Table 1,gender was a categorical variable, which was expressed as frequencies and percentages(%) and compared using Likelihood-ratio Chi squared test. The other variables, such as RTS, SI, SS, MAP, RR, etc., were continuous variables, which were tested for normality using Shapiro–Wilk test. All of the continuous variables in the current study, failing to conform to normality, were thus expressed as median (inter quartile range, IQR) and compared using Mann-Whitney test. The results obtained by SS=SI/SpO2, have fractional parts, so it may be more appropriate to treat it as continuous variable. P value indicated whether there was statistical difference between the two groups in sex, age, RTS, SI, SPO2, SS, MAP, T, and so on. 

Comment 4: In order for this index to be relevant, it is not enough to present the sensitivities and specificities. One should construct a table 3 with different cutoffs for SS in which the proportion of patients dying and surviving is presented for each cutoff value. The authors of the shock index provide us with an optimal cutoff to which we can related when assessing the patients’ prognosis. Why did the authors of this manuscript only offer us an AUC evaluation?

Response: As shown in the ROC curve (Fig 1), each point on the curve corresponds to a sensitivity and specificity, and a high sensitivity means a decrease in specificity. In order to determine an ideal cutoff value, we used the Youdenindex method (Youdenindex= sensitivity + specificity-1), which meaned that we used the SS value corresponding to the maximum value of Youdenindex as its cutoff value. The cutoff value of SS was 1.06 (sensitivity:92.26%, specificity:61.29%, Youdenindex:0.5355). AUC is usually used to evaluate the diagnostic or predictive value of a clinical index, the larger the AUC value, the better the predictive or diagnostic value of the index. As shown in Table 3, we compare the AUC of SS and SI, the results show that the AUC of SS is higher than that of SI and the difference is statistically significant (P= 0.001), indicating that SS is indeed superior to SI and has a certain clinical application value.

---

## [Editor Report · Decision Letter 1]

30 Jun 2020

The ratio of shock index to pulse oxygen saturation predicting mortality of emergency trauma patients

PONE-D-20-08124R1

Dear Dr. Chen,

We’re pleased to inform you that your manuscript has been judged scientifically suitable for publication and will be formally accepted for publication once it meets all outstanding technical requirements.

Kind regards,

Itamar Ashkenazi

Academic Editor

PLOS ONE
---

## [Editor Report · Acceptance letter]

8 Jul 2020

PONE-D-20-08124R1 

The ratio of shock index to pulse oxygen saturation predicting mortality of emergency trauma patients 

Dear Dr. Chen:

I'm pleased to inform you that your manuscript has been deemed suitable for publication in PLOS ONE. Congratulations! Your manuscript is now with our production department. 

Kind regards, 

on behalf of

Dr. Itamar Ashkenazi 

Academic Editor

PLOS ONE